# Dual Tunable Electromagnetically Induced Transparency Based on a Grating-Assisted Double-Layer Graphene Hybrid Structure at Terahertz Frequencies

**DOI:** 10.3390/nano12213853

**Published:** 2022-11-01

**Authors:** Xu Zhong, Tiesheng Wu, Zhihui Liu, Dan Yang, Zuning Yang, Rui Liu, Yan Liu, Junyi Wang

**Affiliations:** 1Guangxi Key Laboratory of Wireless Broadband Communication and Signal Processing, School of Information and Communication, Guilin University of Electronic Technology, Guilin 541004, China; 2Key Laboratory of Optoelectronic Devices and Systems of Ministry of Education and Guangdong Province, College of Optoelectronic Engineering, Shenzhen University, Shenzhen 518060, China; 3Guangdong and Hong Kong Joint Research Centre for Optical Fiber Sensors, College of Optoelectronic Engineering, Shenzhen University, Shenzhen 518060, China

**Keywords:** grating, graphene, EIT-like, subwavelength structures

## Abstract

We propose a graphene plasmonic structure by applying two graphene layers mingled with a thin gold layer in a silicon grating. By utilizing the finite-difference time-domain (FDTD) method, we investigate the optical response of the system, and observe that the design achieves dual tunable electromagnetically induced transparency (EIT)-like effect at terahertz frequencies. The EIT-like effect arises from the destructive interference between the grapheme-layer bright modes and the gold-layer dark mode. The EIT-like phenomenon can be adjusted by the Fermi level, which is related to the applied voltage. The results show that the group delay of the present structure reaches 0.62 ps in the terahertz band, the group refractive index exceeds 1200, the maximum delay-bandwidth product is 0.972, and the EIT-like peak frequency transmittance is up to 0.89. This indicates that the device has good slow light performance. The proposed structure might enable promising applications in slow-light devices.

## 1. Introduction

Electronically induced transparency (EIT) is a quantum interference phenomenon that produces strong dispersion within the transparency window, which leads to slow light effects and enhanced nonlinear magnetization. In 1991, Harris et al. first observed and proposed EIT [1], but the extreme conditions for achieving quantum EIT limit its development and applications. To overcome the limitation of conventional EIT conditions, researchers have studied the EIT-like phenomenon in other systems, such as optical microcavities, optical wave-guides, metamaterials, etc. The EIT-like phenomenon generated by metamaterials, i.e., plasma-induced transparency (PIT), has attracted great interest among researchers. In 2009, Chinese researcher Liu et al. published an exploratory paper in Nature Materials on the interaction of metal dipole and quadrupole patterns to create transparency [2]. In 2018, Yu et al. proposed a planar metamaterial structure with open resonant rings combined with metal strips to achieve EIT-like effects [3]. In the same year, Jia et al. proposed a metasurface consisting of two identical and orthogonal double-ended half-ring metal resonators by numerical simulation to realize the polarization-independent EIT-like effect [4]. In 2019, Han et al. achieved a high-quality-factor EIT-like effect by introducing FP resonators in dielectric guided-mode resonant systems [5]. In general, EIT-like effects can be excited in two ways, one is destructive interference generation between bright and dark modes, and the other is resonant excitation of trap modes in planar metamaterials based on structural symmetry breaking. As far as we know, although EIT-like effects have been achieved in some nanostructures, the EIT-like effect is modulated by carefully adjusting the structural parameters, making the device applicability limited.

Graphene, a two-dimensional honeycomb lattice structure material, has caused great interest in optoelectronic applications due to its high electron mobility and unique doping ability [6,7]. With the study of its material properties, it has been found that graphene has optical properties similar to those of noble metals in the terahertz band and can excite surface plasmas at specific wavelengths. Still, it has some incomparable advantages over noble metals. For example, it has a lower ohmic loss due to low absorption of light in a wide range of frequencies [8]; on the other hand, it can control the Fermi energy level of graphene by electron doping or chemical doping artificially, and the chemical potential can be adjusted directly by adjusting the applied voltage without changing the micro-nano models. This excellent property of tunable and low loss makes it potentially valuable for applications such as photoelectric detection [9] and light modulation [10]. In addition to tuning by doping and applied voltage, graphene can be tuned by changing other parameters. For example, electrical conductivity and carrier concentration are electrically modulated. Kalhor et al. used this tunability to further control THz transmission, group delay, and the EIT response [11]. Nam et al. verified that the absorption behavior of srr could be inhibited by increasing the resistance of graphene ink [12]. Michal et al. proposed that adjusting the dielectric thickness between graphene layers could also change its photoelectric effect [13]. Based on this characteristic, in recent years, many researchers have applied graphene to design nanostructures, and proposed a series of tunable EIT-like devices. For example, in 2020, Xiao et al. proposed a graphene subsurface structure. The design has a group delay of 0.15 ps and a group refractive index of 200 when the chemical potential is set as 0.8 eV [14]. In 2021, Cao et al. investigated a planar terahertz-like EIT metamaterial consisting of a single-layer graphene micro-ring resonator and a single-layer graphene microstrip resonator with a group delay of 1.49 ps and a group refractive index of 400 due to the large thickness [15]. In the same year, Jia et al. demonstrated an EIT-like metasurface that could be used as a refractive index sensor [16], and Tang et al. showed a graphene-like EIT metasurface with the ability of polarization control, in which the group delays of polarized light in x- and y-directions can reach 0.06 ps and 0.08 ps, respectively [17]. Using ultrafast optical-pump THz-probe spectroscopy, Fu et al. investigated the negative terahertz photoconductivity phenomenon based on a photoexcited graphene structure [18]. The device can achieve a modulation depth of 18.8% with a pump power of 200 mW. Torres et al. demonstrated a fascinating optoelectronic switching effect and nonlinear enhancement transmission based on an array structure [19]. Obviously, graphene has been proven to be a promising material for EIT-like systems due to its remarkable characteristics.

In this work, we applied two graphene layers and a thin gold layer to a silicon grating to achieve a tunable EIT-like effect at terahertz wavelengths. In the proposed structure, the upper and lowermost graphene layers are directly excited by the incident light and act as bright modes, while the middle gold layer cannot be directly excited, and a local strong resonance effect between the dark and bright modes occurs, thus the gold layer is considered a dark mode. The destructive interference between the bright and dark modes resulted in plasma-induced transparency. We utilized the finite difference time domain (FDTD) method to analyze its transmission characteristics in detail. The results show that the present structure can achieve a double EIT-like phenomenon. The proposed design can achieve a group delay above 0.6 ps, and the maximum group refractive index reaches 1240. The peak transmittance exceeds 0.89 for the range of 1–10 THz. These merits, including the big group delay, the high group refractive index, big delay-bandwidth product, high transmission efficiency, and good tuning effect, render the proposed structure a good candidate for slow-light devices.

## 2. Structure and Theoretical Model

A unit of the designed period structure is shown in Figure 1, where (a) is the schematic diagram of the three-dimensional structure and (b) is the cross-section view of it. The design consists of two graphene layers and a gold layer sandwiched in silicon grating. More precisely, the upper graphene layer passes through the air grating and is inserted into the silicon grating at both ends, and the middle Au layer lies only in the middle of the dielectric silicon, and the lower graphene layer passes through the air grating but is not inserted into the silicon grating at both ends. By epitaxially extending the graphene layer, the chemical potential of graphene in the structure can be adjusted by applying an applied voltage. We assume that the device extends semi-infinitely along the positive y-direction and that the graphene layer is separately epitaxialized in the negative y-direction to connect it to the electrode. The structure parameters of the proposed design are as follows: Period P = 1000 nm, height H = 150 nm, the width of the upper layer of graphene w1 = 920 nm, the width of the central gold layer w2 = 300 nm, the width of the underlying layer of graphene w3 = 700 nm, h1 = 25 nm, h2 = 50 nm, h3 = 50 nm, h4 = 25 nm, the thickness of the thin gold layer in the middle h = 16nm. Since the device operates in the terahertz region at room temperature, we use the Drude-like model to characterize the electrical conductivity of graphene [20]:(1)$$\sigma =\frac{ie^{2}E_{F}}{\pi \hbar^{2}\left(\omega +i\tau^{-1}\right)}$$
where this simplified essential condition is $E_{F}\gg \left(\hbar \omega ,k_{B}T\right)$ in the near-infrared to THz region when the temperature is set as T = 300 K. In the formula, $E_{F}$ is the Fermi level, ℏ is the reduced Planck constant, ω is the angular frequency, $k_{B}$ is the Boltzmann constant, i is the imaginary unit, and e and τ are the elementary charge and carrier relaxation time, respectively. It is worth noting that the carrier relaxation time also satisfies the following equation: $\tau =\mu E_{F}/\left(e\nu_{F}{}^{2}\right)$, where the carrier mobility is 3.00 m^2^/(V·s) and the Fermi velocity $\nu_{F}$ is 10^6^ m/s [13]. The Fermi energy level of monolayer graphene can be dynamically adjusted in the range of 0.2 eV to 1.2 eV under the application of appropriate bias voltage [21,22,23,24]. Therefore, the Fermi energy level of the simulation is set between 0.6 eV and 0.7 eV in this paper. We numerically investigate the transmission characteristics of the presented structure using the finite difference time domain (FDTD) method. In the simulation, the dielectric constant of silicon is set as $\varepsilon_{Si}=11.9$ [25], the dielectric constant of air is 1.0, and the dielectric constant of Au is taken from the Ordal model [26]. The TM polarization wave is incident along the negative z-direction. The periodic boundary conditions are set in the x-direction, and perfectly matched layers are set in the z-direction. Regarding the structure’s infinite extension along the y-direction, we simplify the model to a two-dimensional simulation to reduce the simulation time. The maximum mesh steps at the x- and z-directions are set as Δx=2 nm and Δz=1 nm, respectively.

## 3. Results and Discussion

For comparison, as shown in Figure 2a, we numerically investigate the transmission spectra of the silicon grating with three layers of materials (solid black line), only upper-layer graphene (red dotted line), only middle-layer thin-film gold (blue dotted line), and only lower-layer graphene (green dotted line) when the graphene chemical potential is set as 0.6 eV. From Figure 2a, we can be seen that both the upper and lower layers of graphene are directly coupled with the incident light acting as bright modes, and the middle layer of gold is not directly coupled to light, acting as a dark mode. The solid black line is the transmission spectrum obtained by the device when all three layers of graphene and gold are present. It can be seen that the destructive interference between the two bright modes and one dark mode produces two EIT-like transmission peaks. Figure 2b not only plots the transmission curve but also plots the reflection and absorption curves. It can be seen that this structure has three reflection peaks, and the absorption rate is more prominent at the two frequency positions, mainly due to the absorption loss caused by gold. According to the transmission characteristics in Figure 2b, the peak transmittance of the two EIT-like transmission peaks is higher than 0.6, and the transmittance of the frequency transmission peak can reach 0.89. Considering the light absorption rate of each layer of graphene and gold thin layer during interference, the light transmittance of this structure is good. Figure 3a,c present the electric field at the spectral dip (4.34 and 7.40 THz, respectively) when the upper and lower graphene layers are present alone, with significant electric field enhancement at both ends of the graphene. When only the middle gold layer is present, the electric field diagram is shown in Figure 3b. With no significant electric field enhancement effect throughout the simulation band, the electric field diagram of the device at 3.43 THz is given as a demonstration in the figure. This further illustrates that the upper and lower graphene can be effectively excited by incident light and act as bright modes, while the middle layer of thin gold cannot be effectively excited by incident light and acts as the dark mode.

The electric field distributions of the structure at the three dips and two peaks are given in Figure 4a–e. It can be seen that the first dip is generated by the destructive interference between the upper-layer graphene and middle-layer gold, which can be understood as light–dark mode coupling, while the second and third ones are generated from the three layers of graphene and gold bright–dark mode coupling. In contrast, the lower-frequency peak is generated by the bright–dark mode coupling of the upper and middle layers of graphene and gold, and the higher-frequency peak is mainly caused by the bright mode of the lower layer of graphene.

In order to explore the tunability of the device, Figure 5a,b show the transmission spectrum and reflection spectrum of the device for different chemical potentials. It can be seen from the figure that when the chemical potential increases from 0.6 eV to 0.7 eV, the transmission spectrum of the device and reflectance spectra are shifted to higher frequencies. In order to observe the changes of each inflection point of the transmission spectrum more clearly, Figure 5c,d show the changing trend of the three dips and two peaks of the transmission spectrum, where dip1, dip2, and dip3 are defined as three transmission valleys from a lower frequency to a higher frequency and peak1 and peak2 are described as three transmission peaks from a lower frequency to a higher frequency. It can be seen that the frequencies of the three dips and the two peaks have an approximately linear relationship with the chemical potential, and the peak value is more sensitive to changes in the chemical potential. In order to show the corresponding relationship between the applied voltage of graphene and the chemical potential, the following equation is given [27]:(2)$$E_{F}=\hbar \nu_{F}\sqrt{\frac{\pi \varepsilon_{r}\varepsilon_{0}V_{g}}{ed}}$$
where the $\varepsilon_{r}$ and $\varepsilon_{0}$ represent the permittivity of the insulator layer and vacuum, respectively, d is the insulator layer thickness, and $V_{g}$ is the applied voltage. Therefore, the chemical potential of graphene can be adjusted by applying an applied voltage, thereby altering the device application frequency.

The slow-light effect is an important phenomenon in EIT-like devices, and the performance is mainly measured by the group delay ($t_{g}$) and group refractive index ($n_{g}$), both of which are determined by the following formulae [28,29,30]:(3)$$t_{g}=-d\varphi (\omega )/d\omega$$
(4)$$n_{g}=(c/h)\times t_{g}$$
where φ, ω, c, and h are the transmission phase, angular frequency, light velocity in the vacuum, and the total thickness of the unit cell, respectively. The curves of the phase and group delay obtained from the present structure as a function of frequency during the growth of the graphene chemical potential from 0.6 eV to 0.7 eV in 0.025 eV increments are given in Figure 6a–e, and it is known that the present device is capable of achieving a stable 0.6 ps group delay within the proposed chemical potential interval. Figure 7 presents the group delay and the group refractive index with the change in chemical potential; it can be more intuitively seen from the figure that the group refractive index of this device can reach 1240 when the group delay reaches 0.62 ps. Furthermore, in order to compare the results of this design with other research models more intuitively, Table 1 lists the slow-light effect results obtained by the reference and the model in this paper. It can be seen that the group refractive index of the design is better than most reference data mentioned.

For the slow-light effect, an important characteristic parameter is the delay-bandwidth product (DBP) [31,32]. Devices with large DBP play an important role in communication. The value of DBP is determined by the following equation:(5)$$DBP=\Delta f\times t_{g}$$
where Δf represents the full width at half maximum (FWHM) of the EIT-like peak and $t_{g}$ is the maximum group delay. Since the mean value of FWHM in the range of 0.6 eV to 0.7 eV for the two EIT-like peaks of the transmission spectrum obtained from this device are 1.62 THz and 0.46 THz, the two types of EIT effect DBP are 0.972 and 0.276, respectively.

**Table 1 nanomaterials-12-03853-t001:** Comparison of the proposed structure and the latest relevant designs.

Reference	Polarization Dependence	Main Material	Tunable	*t_g_* (ps)	$n_{g}$
[3]	Yes	Ag	No	N/A	N/A
[5]	Yes	CaF_2_, Si	No	N/A	N/A
[14]	Yes	Graphene, CaF_2_	Yes	0.15	200
[17]	No	Graphene	Yes	0.08	240
[33]	No	Copper	No	N/A	64
[34]	Yes	Au	No	2	29.7
This work	Yes	Graphene, Si	Yes	0.62	1240

The effect of graphene carrier mobility on the slow-light effect is explored in Figure 8. From Figure 8a, it can be seen that the increase in carrier mobility makes the variation of the structural phase at the inflection point stronger. According to formula (2), it can be seen that the change rate of the phase will directly affect the group delay of the structure. The group delay increases with increasing carrier mobility, which is consistent with the changing trend of the curve shown in Figure 8b. From formula (3), we further obtain the group refractive index curve in Figure 8c. It is observed that the increase in the carrier mobility causes the group refractive index of this design to exceed 1240, and the slow-light effect is significantly improved.

Figure 9 explores the effect of replacing the thin gold layer in the middle of the structure with other metals on the EIT-like effect. Obviously, the transmission spectra of gold and silver are basically the same, but the use of iron, platinum, titanium, etc., will weaken the EIT-like effects of the transmission spectra to varying degrees. Because the metal in this structure has two sides that will be in direct contact with the air, considering the silver in the air easily oxidizes, we chose gold as the middle-layer metal material.

Figure 10 shows the transmission and group delay spectra of graphene-only and graphene–gold structures when the middle layer of gold is replaced with graphene. It can be seen from the figure that compared with the graphene-only structure, the value of the high-frequency peak of the graphene–gold structure is significantly reduced, and both peak points have frequency shifts. However, the group delay spectrum shows that the introduction of gold increases the maximum group delay of this structure by approximately 0.15 ps, and the slow-light effect is better.

In order to illustrate the determination process of the structural parameters, Figure 11 shows the influence on the transmission spectrum and the group delay spectrum when the width of the upper and lower graphene layers change. It can be seen from Figure 11a,b that the increase in the width of the upper graphene has little effect on the peak value and position of the first transmission peak, but the peak value of the second transmission peak increases with the increase in the graphene width. The time-lapse spectra show that as the upper graphene width increases, the maximum group delay first increases and then decreases. Since the purpose of this design is to obtain a device structure with high group delay, this design has potential application value in the slow-light effect, so the upper graphene width is determined to be the value of the maximum group delay point, that is, 920 nm. Under the condition that the width of the upper graphene layer is kept unchanged (that is, the selected optimal value is 920 nm), Figure 11c,d presents the transmission spectrum and the group delay spectrum of the change in the width of the lower graphene layer. It can be seen that the change in the width of the lower layer graphene will affect the peak value and position of the second transmission peak significantly, and the group delay spectrum also increases first and then decreases with the increase in the width of the underlying graphene. In order to obtain a high group delay, the width of the underlying graphene is determined to be 700 nm.

We further discuss the influence of structural parameter changes on the transmission results of this structure. Figure 12a,b respectively show the transmittance scans obtained by this design when the thickness and width of the thin gold layer in the middle of the structure change. It can be seen from Figure 12a that when the thickness of the thin gold layer increases to more than 13 nm, dip1, dip2, and peak1 all display the red-shift phenomenon, as peak2 is red-shifted and the intensity increases. Figure 12b shows that when the width of the gold thin layer increases, dip1 becomes narrow, peak1 and dip1 are red-shifted, peak2 is slightly red-shifted while the intensity increases, and dip3 does not change much. The scanning results are in good agreement with the correlation between graphene and the thin gold layer at the peak point shown by the electric field map. Therefore, the transmission characteristics can be tuned by adjusting the parameters of the thin gold layer.

Finally, we investigate the effect of the polarization state of incident light on the EIT-like effect of this structure. Figure 13 shows the variation in the transmission efficiency with the polarization angle of the incident light. At a frequency range of 1–10 Hz, as the TM-polarized wave gradually turns into a TE-polarized wave, we can see that the transmission efficiency decreases with an increasing polarization angle. When the polarization angle reaches approximately 75°, the entire transmittance spectrum essentially drops to 0. At the same time, the group delay and the group refractive index decrease gradually because the phase change is smaller and smaller. When the TM wave is completely transformed into the TE wave, both the group delay and group refractive index are approximately equal to 0. The EIT effect basically disappears. It can be seen that in the range of 1–10 THz, the device is relatively sensitive to the polarization mode of the incident light, and the TE polarized wave cannot pass through the structure with high transmittance.

## 4. Conclusions

In summary, we propose and numerically demonstrate a dynamically tunable EIT-like device in the terahertz range. The proposed structure is made of two layers of graphene, one layer of gold film, and a silicon grating. The two layers of graphene can strongly couple to the incident electromagnetic wave under the excitation of the periodic grating. The layers of graphene and gold operating in bright and dark modes, respectively, coupled with each other to produce two EIT-like transmission peaks. The spectral resonance position can be tuned by adjusting the chemical potential of graphene. The device utilizes a skinny thickness to achieve a large group delay while maintaining high transmittance. The slow light effect of the design is superior to most known similar structures. The maximum group delay reaches 0.62 ps, the group refractive index is 1240, the delay-bandwidth product reaches 0.972, and the maximum peak transmittance exceeds 0.89. The proposed structure has good prospects for applications in the fields of slow light, switching, and communication.

## Figures and Tables

**Figure 1 nanomaterials-12-03853-f001:**
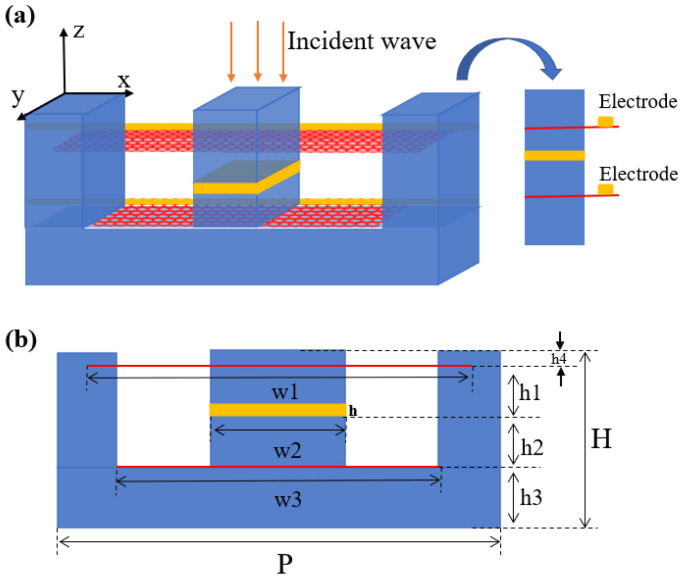
Schematic illustration of the proposed structure: (**a**) The whole schematic; (**b**) profile view.

**Figure 2 nanomaterials-12-03853-f002:**
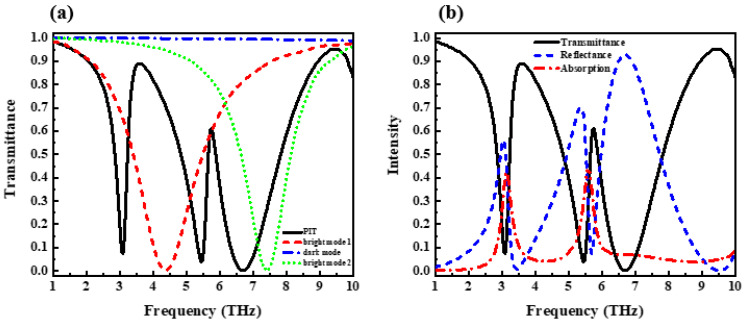
(**a**) Transmission spectrum in different modes. (**b**) Transmission characteristics of this design.

**Figure 3 nanomaterials-12-03853-f003:**
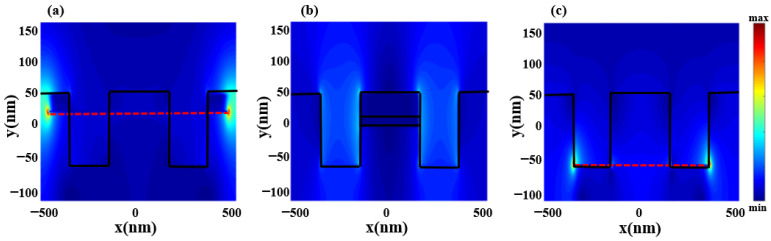
Distribution of the electric field when two layers of graphene and a thin layer of gold are present separately for this structure. (**a**) Upper layer graphene at resonant frequency. (**b**) Mid-layer of gold without resonance. (**c**) Lower layer graphene at the resonant frequency.

**Figure 4 nanomaterials-12-03853-f004:**
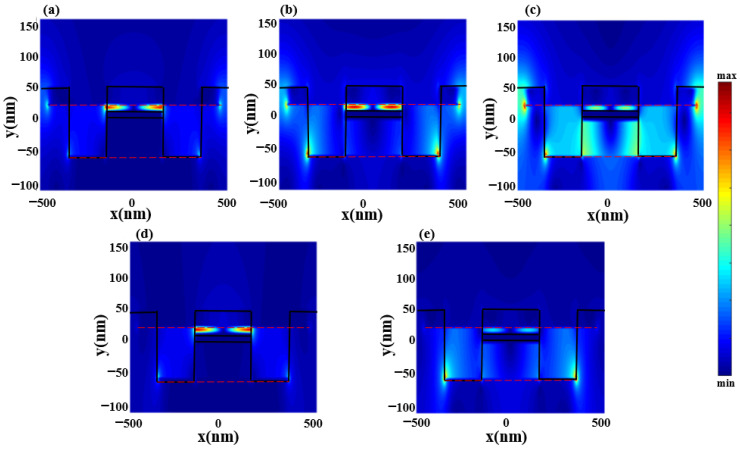
Distribution of the electric field at frequencies of (**a**) 3.08 THz, (**b**) 5.46 THz, (**c**) 6.69 THz, (**d**) 3.59 THz, and (**e**) 5.75 THz.

**Figure 5 nanomaterials-12-03853-f005:**
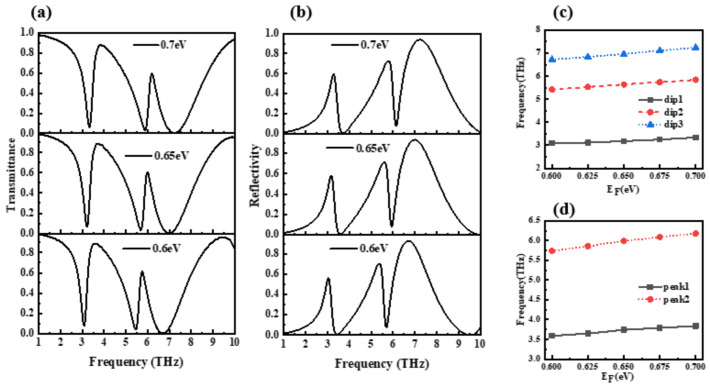
Variation of transmission characteristics with chemical potential. (**a**) Variation of transmission spectrum with chemical potential. (**b**) Variation of reflection spectrum with chemical potential. (**c**) Trend of transmission spectrum dips. (**d**) Trend of transmission spectrum peaks.

**Figure 6 nanomaterials-12-03853-f006:**
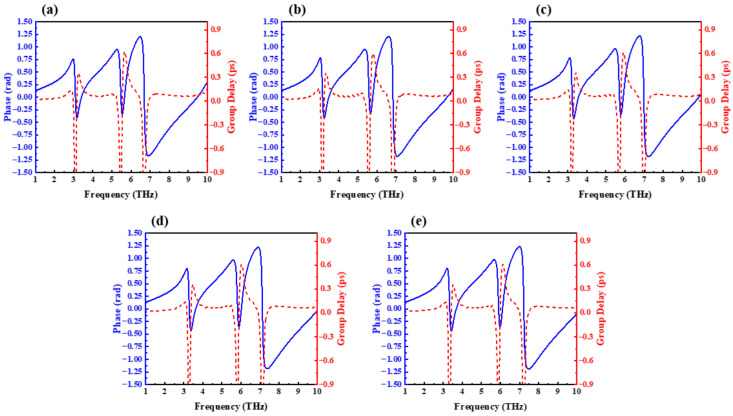
Phase curve and group delay curve with different chemical potential. (**a**) 0.6 eV, (**b**) 0.625 eV, (**c**) 0.65 eV, (**d**) 0.675 eV, and (**e**) 0.7 ev.

**Figure 7 nanomaterials-12-03853-f007:**
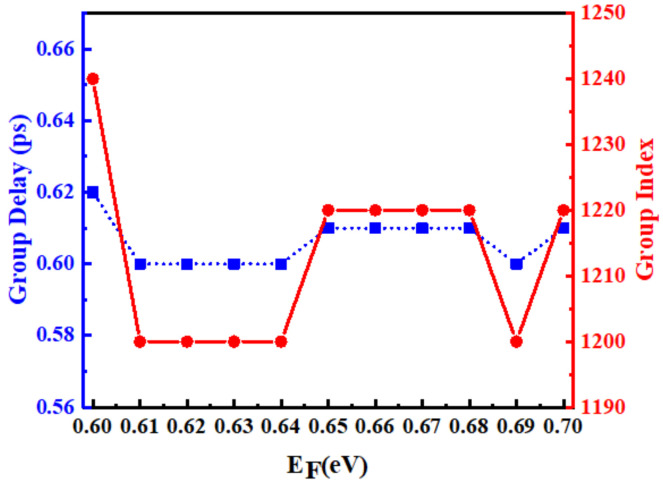
Variation of group delay and group index with chemical potential.

**Figure 8 nanomaterials-12-03853-f008:**
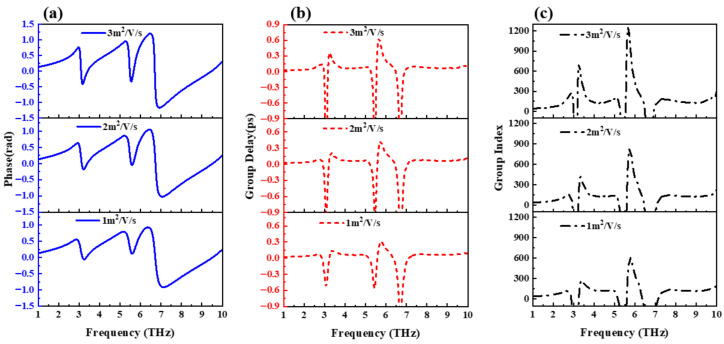
Influence of graphene’s carrier mobility on structural slow-light effect. (**a**) Effect on phase. (**b**) Effect on group delay. (**c**) Effect on group index.

**Figure 9 nanomaterials-12-03853-f009:**
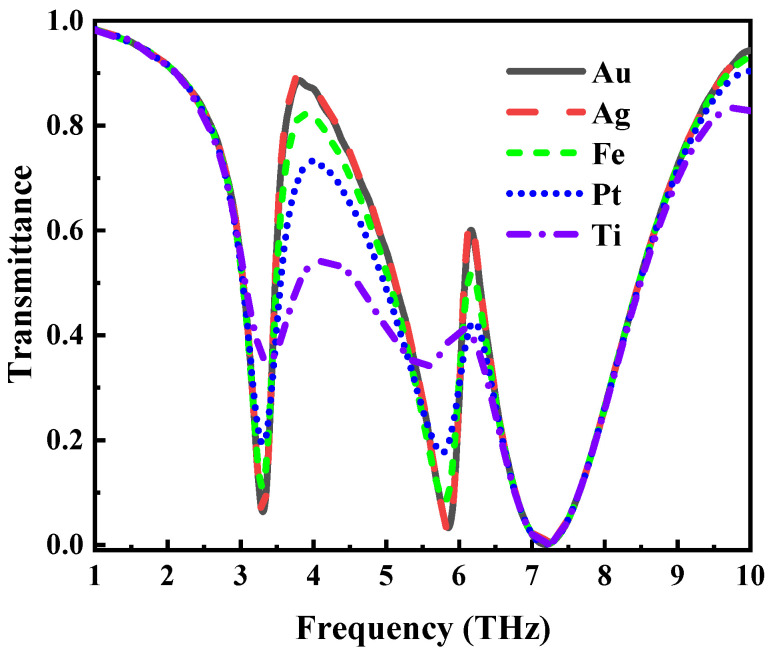
Changes in the transmission spectrum of metal replacement in this structure.

**Figure 10 nanomaterials-12-03853-f010:**
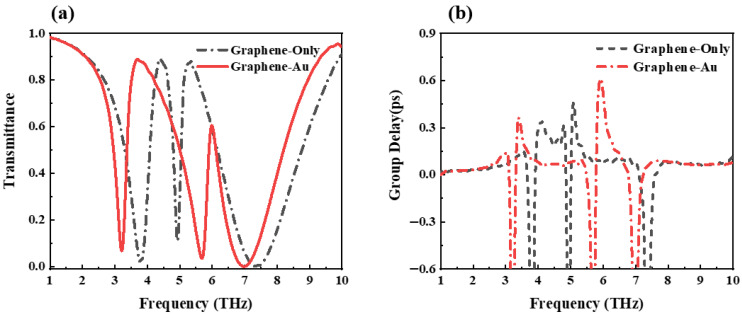
Comparison of (**a**) transmission spectrum and (**b**) group delay spectrum obtained by graphene-only structure and graphene–gold structure.

**Figure 11 nanomaterials-12-03853-f011:**
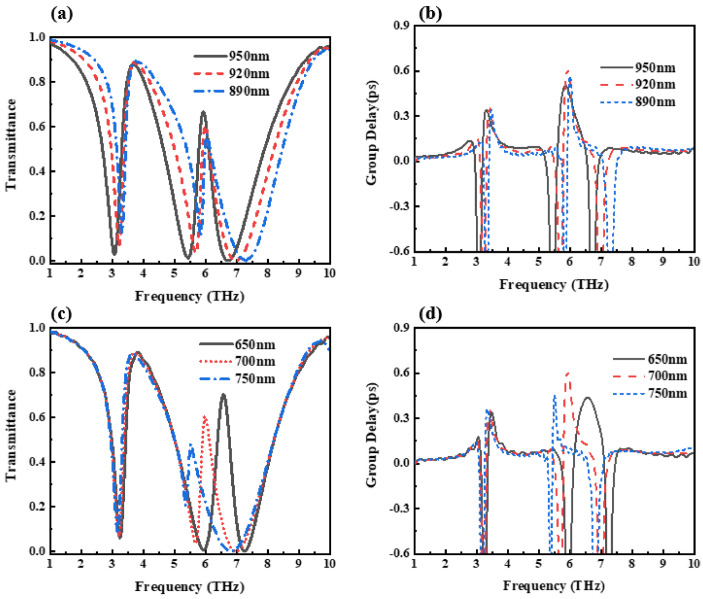
The effect of the upper and lower graphene width variation on the EIT effect. (**a**) The structure transmission spectrum. (**b**) Group delay change when the width of the upper graphene layer changes. (**c**) The structure transmission spectrum. (**d**) Group delay change when the width of the lower layer graphene changes.

**Figure 12 nanomaterials-12-03853-f012:**
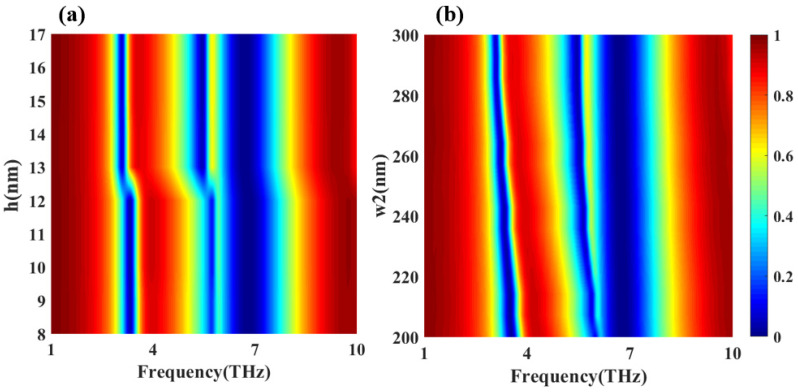
(**a**) The transmission efficiency as a function of the thin gold layer thickness for the frequency range of 1–10 THz. (**b**) The transmission efficiency as a function of the thin gold layer width for the frequency range of 1–10 THz.

**Figure 13 nanomaterials-12-03853-f013:**
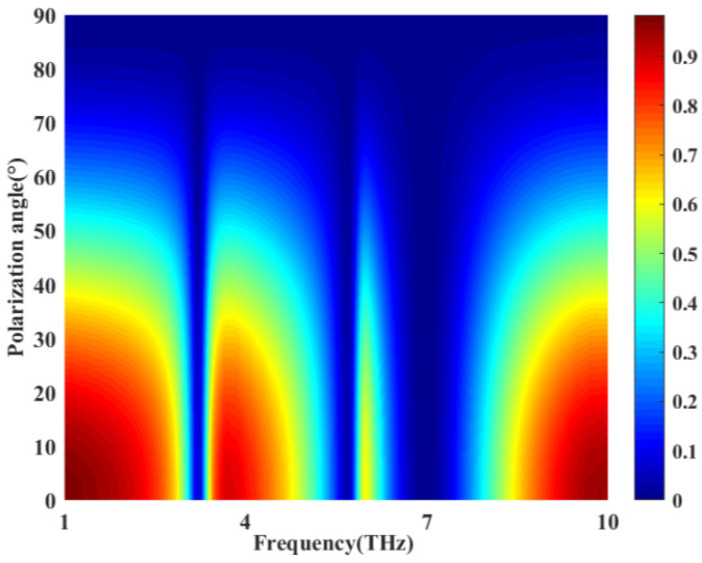
The transmission efficiency of the proposed structure for different polarization angles in the frequency range of 1–10 THz.

## Data Availability

The data presented in this study are available upon reasonable request from the corresponding author.

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
