# Peer review of "Dual Tunable Electromagnetically Induced Transparency Based on a Grating-Assisted Double-Layer Graphene Hybrid Structure at Terahertz Frequencies"

_nanomaterials, 2022, doi:10.3390/nano12213853_

Round 1
Reviewer 1 Report (Previous Reviewer 2)
The authors have clarified the most important issues previously raised in the review stage. The results are interesting and the conclusions are solid. Then, in my opinion, this work can be considered for publication in present form
Author Response
Dear Reviewer:
Thanks very much for your kind work and consideration on publication of our paper. On behalf of my co-authors, we would like to express our great appreciation to editor and reviewers.
Thank you and best regards.
Reviewer 2 Report (New Reviewer)
The reviewed article concerns the theoretical concept of dual tunable electromagnetically induced transparency effect coupling in a grating-assisted graphene-Au metamaterial. The manuscript is quite interesting, although the physics of such phenomena is not new and has been described in numerous works. Nevertheless, I am asking the authors to carefully refer to the following issues:
1. In the abstract, a few sentences are written in red. Not only are repetitions there, but also the language itself and the vocabulary used is very poor. In general, English should be thoroughly examined throughout the manuscript. Even the title of the manuscript itself seems a bit confusing and is not fully written correctly.
2. In the introduction (lines 63 and 64), the authors mention several applications of the graphene-based concepts. However, they overlooked several works, e.g .: Scientific Reports 11.1 (2021): 1-11; Crystals 12.4 (2022): 525; Nanomaterials 11.11 (2021): 2999.
3. Figure 1 (a) is unclear to me. The schematic diagram of the device suggests that the top layer of graphene is behind it. In addition, I suggest describing the individual layers/parts by adding text. Finally, the markings are illogical. In my opinion, H should equal h1+h2+h3+h4 and W should equal w1+w2+w3…But it's not like that……
4. The authors define tunability by changing the Fermi energy level of graphene. However, for greater clarity, I suggest to the authors to specify what voltage should be applied to obtain a given Fermi energy level.
5. The effect of coupling the resonance modes is very slight in Figure 4. Therefore, the question arises whether the geometrical parameters of the proposed structure were properly optimized?
Author Response
Dear Editors and Reviewers:
On behalf of my co-authors, we thank you very much for giving us an opportunity to revise our manuscript, we appreciate the editor and reviewers very much for their positive and constructive comments and suggestions on our manuscript entitled “Dual tunable electromagnetically induced transparency effect coupling in a grating-assisted graphene-Au metamaterial structure" (ID: nanomaterials-1904105). Those comments are all valuable and very helpful for revising and improving our paper, as well as the important guiding significance to our research. We have studied the comments carefully and have made a correction which we hope meets with approval. We made a lot of changes to make our article more complete, and revised portions are marked in red on the paper. The responses to the Reviewer's comments are as follows:
- Response to comment: (In the abstract, a few sentences are written in red. Not only are repetitions there, but also the language itself and the vocabulary used is very poor. In general, English should be thoroughly examined throughout the manuscript. Even the title of the manuscript itself seems a bit confusing and is not fully written correctly.)
Response: According to the reviewer's comments, we have revised the relevant content of the paper, including the title, abstract and main text.
- Response to comment: (In the introduction (lines 63 and 64), the authors mention several applications of the graphene-based concepts. However, they overlooked several works, e.g .: Scientific Reports 11.1 (2021): 1-11; Crystals 12.4 (2022): 525; Nanomaterials 11.11 (2021): 2999.)
Response: Considering the Reviewer's suggestion, we have added references and related statements on lines 62 to 68, highlighted in red.
- Response to comment: (Figure 1 (a) is unclear to me. The schematic diagram of the device suggests that the top layer of graphene is behind it. In addition, I suggest describing the individual layers/parts by adding text. Finally, the markings are illogical. In my opinion, H should equal h1+h2+h3+h4 and W should equal w1+w2+w3…But it's not like that……)
Response: According to the Reviewer's suggestion, we have made some changes to the notation of the structural parameters. However, due to the layered structure of this device, the width of each sandwich cannot be directly added or subtracted on a horizontal plane, so we changed the width of the device to a more accurate period P, but the width of graphene layer and metal thin layer is still represented by w. At the same time, some structural symbols have been added to the article, which have been marked in red font.
- Response to comment: (The authors define tunability by changing the Fermi energy level of graphene. However, for greater clarity, I suggest to the authors to specify what voltage should be applied to obtain a given Fermi energy level.)
Response: It is really true as the Reviewer suggested, we added the corresponding equation between the applied voltage and the chemical potential of graphene in this paper (Formula 2). It can be seen that the applied voltage is positively proportional to the chemical potential of graphene.
- Response to comment: (Therefore, the question arises whether the geometrical parameters of the proposed structure were properly optimized?)
Response: Thanks to the reviewers for your suggestions. In fact, our structure is a more optimized result obtained by combining transmittance, group delay and group refractive index after parametric scanning, and further optimization is not very likely.
We tried our best to improve the manuscript and made some changes in the manuscript. These changes will not influence the content and framework of the paper. We appreciate for Editors/Reviewers' warm work earnestly and hope that the correction will meet with approval.
Once again, thank you very much for your comments and suggestions.
Round 2
Reviewer 2 Report (New Reviewer)
The authors took into account and responded in detail to my comments. The manuscript may be published as is.
Author Response
Dear Reviewer:
Thanks very much for your kind work and consideration on publication of our paper. On behalf of my co-authors, we would like to express our great appreciation to editor and reviewers.
Thank you and best regards.
This manuscript is a resubmission of an earlier submission. The following is a list of the peer review reports and author responses from that submission.
Round 1
Reviewer 1 Report
The paper reports on modelling of electromagnetically induced transparency in a grating-assisted structure. In principle, the topic is interesting. However, even at the stage of design it is clear that the paper should be rejected as dealing with a completely unrealistic structure. How it is possible to bury two layers of graphene and one layer of gold somewhere in bulk of silicon grating? Apart from this, gating of graphene means applied voltage through some electrodes, which provides more tricky questions about consistency of the design in general. I think that such structure can be an interesting example for numerical exercises, but its feasibility is at vanishing level, at least at the current level of nanofabrication.
Author Response
Dear Editors and Reviewers:
Thank you for your letter and for the Reviewer's comments concerning our manuscript entitled "Dual tunable electromagnetically induced transparency effect coupling in a grating-assisted graphene-Au metamaterial structure" (ID: nanomaterials-1904105). Those comments are all valuable and very helpful for revising and improving our paper, as well as the important guiding significance to our research. We have studied the comments carefully and have made corrections which we hope meet with approval. Revised portions are marked in red on the paper. The responses to the reviewer’s comments are as flowing:
Responds to the Reviewer's comments:
- Response to comment: (How it is possible to bury two layers of graphene and one layer of gold somewhere in bulk of silicon grating, such structure can be an interesting example for numerical exercises, but its feasibility is at vanishing level)
Response: As far as we know, a number of similar designs have been proposed and analyzed in previous reports, such as the structure proposed by Xu et al. (Optics Express, 25 (17), 20780-20790, 2017) and the structure proposed by Zhang et al. (Optics Express, 24(18), 20002, 2016). In particular, Gao et al. proposed a photodetector combining gold grating and graphene (Optics Letters, 43(6), 1399, 2018), and carried out experimental verification to prove that our structure is feasible, although the current processing process is complicated and the cost is relatively expensive.
- Response to comment: (Gating of graphene means applied voltage through some electrodes, which provides more tricky questions about consistency of the design in general.)
Response: The advantage of using graphene is realizing the electricity adjustment. Regarding the issue of the applied voltage to graphene, there are many references. For example, Huang et al. directly applied voltage to graphene (Optics Express, 22(24), 30108-30117, 2014), while D. LEGRAND et al., similar to this paper, applied voltage to the metal contacted by graphene to regulate graphite (Optics Express, 25(15), 17306, 2017).
Once again, special thanks to you for your comments.
Reviewer 2 Report
The manuscript nanomaterials-1904105 has been devoted to mainly present numerical studies about electromagnetically induced transparency effects exhibited by a particular grating-assisted graphene-Au metamaterial structure. Please see below a list of comments to the authors:
- My main concern is the connection of the results that requires a better presentation in order to see that these results are systematic instead of incidental. The optimization of the parameters in the design and the variation of the parameters for the main observations would allow seeing how these parameters were selected, and how this report can be useful for future research.
- How the authors selected the materials involved in the proposed system over other chemical elements? The main advantages and disadvantages should be highlighted.
- The results of table 1 should be better described in order to see the value of the main results.
- Do the system is dependent on incident polarization?
- An induced photocurrent or photoconductivity can be present in both graphene and Au thin films under electromagnetic irradiation; how these conditions can represent an influence to the main findings here reported? The authors are invited to discuss about this potential issue and see for instance: http://dx.doi.org/10.1080/09500340.2014.943313
https://doi.org/10.1016/j.optcom.2017.01.045
- Figure 5 does not show reflection spectrum as stated.
- From the introduction is not clear how this proposed system holds the definition of metamaterial with or without a negative refractive index for THz. Moreover, how is the contrast in the performance of only graphene or only Au in the same task proposed for the integrated system of Au+graphene analyzed?
- Regarding that no perspectives are discussed, it is not clear the potential impact of the work.
- The keywords should be improved,
- Besides several references could be updated, it is suggested to split the collective references in order to better justify them by individual expression for each citation presented in individual form.
Author Response
Dear Editors and Reviewers:
On behalf of my co-authors, we thank you very much for giving us an opportunity to revise our manuscript, we appreciate the editor and reviewers very much for their positive and constructive comments and suggestions on our manuscript entitled “Dual tunable electromagnetically induced transparency effect coupling in a grating-assisted graphene-Au metamaterial structure" (ID: nanomaterials-1904105). Those comments are all valuable and very helpful for revising and improving our paper, as well as the important guiding significance to our research. We have studied the comments carefully and have made a correction which we hope meets with approval. Revised portions are marked in red on the paper. The responses to the Reviewer's comments are as follows:
- Response to comment: (The optimization of the parameters in the design and the variation of the parameters for the main observations would allow seeing how these parameters were selected, and how this report can be useful for future research.)
Response: As the Reviewer suggested, we supplement the effect of the graphene structure parameter sweep on the simulation results in the article (Fig. 11), proving that our parameter determination is systematic and the result after optimization.
- Response to comment: (How the authors selected the materials involved in the proposed system over other chemical elements? The main advantages and disadvantages should be highlighted.)
Response: Considering the Reviewer's suggestion, we have plotted the transmission spectrum after replacing gold with other metals (Fig. 9) and explained why we chose gold from the spectral results and the actual situation.
- Response to comment: (The results of table 1 should be better described in order to see the value of the main results.)
Response: Table 1 has been supplemented with relevant content.
- Response to comment: (Do the system is dependent on incident polarization?)
Response: It is really true as the Reviewer suggested; the structure should use TM waves as the incident light source. In order to more intuitively reflect the influence of the polarization state of the incident light on the results, a scanning colour map with polarization angle as a variable is added in this paper (Fig. 13). It can be seen that when the incident light gradually becomes a TE wave, the EIT effect gradually disappears.
- Response to comment: (An induced photocurrent or photoconductivity can be present in both graphene and Au thin films under electromagnetic irradiation; how these conditions can represent an influence to the main findings here reported?)
Response: According to the Reviewer's suggestion, we carefully read these two papers, one of which studies the photoconductivity and third-order nonlinear optical effects of gold under laser irradiation, which will increase its transmittance, and the other studies at 0- 1.4THz band, the increase of the pump power lead to a nonlinear increase in the transmittance of graphene. However, the results cannot be directly referenced because the bands used in the two articles are different from those used in this structure. Except, I'm sorry, we haven't mastered the simulation in this area. This is indeed a good research point. In future research, we will definitely carry out research in this area to make it reflected in our future work.
- Response to comment: (Figure 5 does not show reflection spectrum as stated.)
Response: We are very sorry for our negligence in error on the figure, and we have made changes to it.
- Response to comment: (From the introduction is not clear how this proposed system holds the definition of metamaterial with or without a negative refractive index for THz. Moreover,how is the contrast in the performance of only graphene or only Au in the same task proposed for the integrated system of Au + graphene analyzed?)
Response: In fact, in all kinds of micro-nano structures that combine graphene, grating, metal, etc., they are basically defined as metamaterials (Optics Express, 24(18), 20002, 2016; Optics Express, 25 (17), 20780-20790, 2017). We think it is appropriate to define this structure as metamaterials. Moreover, we have added Figure 10 to show the performance comparison of the graphene-only structure and the gold-graphene structure.
- Response to comment: (Regarding that no perspectives are discussed, it is not clear the potential impact of the work.)
Response: Based on the comments of the reviewers, we have made corresponding changes to the introduction, which have been marked in red.
- Response to comment: (The keywords should be improved)
Response: We have changed the keywords based on the Reviewer's suggestion and marked them in red.
- Response to comment: (Besides several references could be updated, it is suggested to split the collective references in order to better justify them by individual expression for each citation presented in individual form.)
Response: We have split and revised the references.
We tried our best to improve the manuscript and made some changes in the manuscript. These changes will not influence the content and framework of the paper. We appreciate for Editors/Reviewers' warm work earnestly and hope that the correction will meet with approval.
Once again, thank you very much for your comments and suggestions.